# Deletion of sulfate transporter SUL1 extends yeast replicative lifespan via reduced PKA signaling instead of decreased sulfate uptake

Juan Long[1,2], Meng Ma[1,2], Yuting Chen[2], Bo Gong[3]*, Yi Zheng[1,2]*, Hao Li[4]*, Jing Yang[1,2]*

[1]Department of Health Management and Institute of Health Management, Sichuan Provincial People's Hospital, University of Electronic Science and Technology of China, Chengdu, China; [2]Laboratory of Aging Research, School of Medicine, University of Electronic Science and Technology of China, Chengdu, China; [3]Department of Laboratory Medicine, Sichuan Academy of Medical Sciences and Sichuan Provincial People's Hospital, School of Medicine, University of Electronic Science and Technology of China, Sichuan, China; [4]Department of Biochemistry and Biophysics, University of California San Francisco, San Francisco, United States

*For correspondence:
gongbo@med.uestc.edu.cn (BG);
yi_zheng@uestc.edu.cn (YZ);
haoli@genome.ucsf.edu (HL);
yangjing1977@uestc.edu.cn (JY)

Competing interest: The authors declare that no competing interests exist.

## eLife Assessment

This study offers a **valuable** contribution to the understanding of how inorganic nutrient transporters, particularly SUL1, influence yeast lifespan through signaling pathways rather than transport functions. The findings suggest a novel link between SUL1 deletion and extended replicative lifespan, supported by transcriptomic and stress–response data. However, the strength of the evidence remains **incomplete**, with key experiments—such as sulfate supplementation tests, functional autophagy validation, and transport assays—either missing or insufficiently described. As a result, while the manuscript presents promising insights, additional work is needed to robustly support its conclusions.

**Abstract** The regulation of cellular metabolism and growth in response to nutrient availability is crucial for cell survival and can significantly impact on lifespan. Central to this regulation is a class of transporters that sense and transport specific nutrients and transduce the signal downstream to control genes responsible for growth and survival. In this study, we identified SUL1, a plasma membrane transporter responsible for regulating the entry of extracellular sulfate in *Saccharomyces cerevisiae*, as a key gene for regulating lifespan. We conducted a systematic analysis to delineate the downstream mechanism underlying the lifespan extension by SUL1 deletion. Surprisingly, we found that the lifespan-extending effect of SUL1 deletion is not due to decreased sulfate transport. The SUL1 deletion mutant exhibited decreased PKA signaling, resulting in a series of downstream effects, including increased stress-protective trehalose and glycogen, increased nuclear translocation of MSN2, elevated expression of general stress response genes, enhanced autophagy, and reduced expression of amino acid biosynthetic and ribosomal genes. We demonstrated that the observed increase in lifespan is dependent on MSN2 and autophagy pathways. Our findings exemplify the influence of nutrient signaling rather than the nutrient itself on lifespan regulation and further substantiate the pivotal role of the PKA pathway in this process.

## Introduction

The metabolism of nutrients is known to have a significant regulatory impact on the physiology and growth of organisms (*González and Hall, 2017*). Reduced food intake has been shown to extend the lifespan across many eukaryotes, such as yeast, worms, flies, and mice (*Carmona and Michan, 2016*; *Green et al., 2022*). The regulation of longevity has been documented to involve distinct routes and regulators, such as the integrated stress response (ISR), AMPK signaling pathway, and mTORC1 signaling pathway. These pathways are activated in response to various nutrient restriction stresses, including glucose, amino acid, and lipid restriction (*Loewith and Hall, 2011*; *Wullschleger et al., 2006*). Eukaryotic cells have evolved distinct mechanisms for nutrition sensing and transportation to exert regulatory control over various cellular targets. The deregulation of nutrient sensing, widely recognized as a hallmark of aging, can influence lifespan and healthspan in a variety of animal models (*Acosta-Rodríguez et al., 2022*; *Duran-Ortiz et al., 2021*; *Fan et al., 2021*).

Recently, several nutrient transporters have been implicated in the mediation of aging or age-related illnesses. For example, the inhibition of glucose transporter SGLT-2 has the potential to delay cellular senescence (*La Grotta et al., 2022*). The amino acid transporter SLC36A4 can regulate the mTORC1 signaling in lysosomes of retinal pigmented epithelial cells through its regulation of the amino acid pool (*Shang et al., 2017*). Apo E, which serves as a constituent of plasma lipoproteins responsible for lipids transportation, plays a role in several neurodegenerative disorders (*Martens et al., 2022*). In addition to organic nutrients, organisms are subject to the regulation of a vast amount of inorganic nutrients including phosphate and sulfate. However, the impact of these nutrients transporters on longevity remains largely unexplored.

The yeast has been widely used as a model organism in the study of molecular mechanisms underlying aging, as well as evolutionarily conserved signal pathways. Yeast cells possess a diverse array of inorganic nutrient sensing systems that facilitate rapid adaptation to fluctuating environments. For instance, the MEP2 system detects ammonium (*Van Nuland et al., 2006*), the PHO84 system senses phosphate (*Giots et al., 2003*), the FTR1 system is involved in iron sensing, the ZRT1 system detects zinc (*Schothorst et al., 2017*), and the SUL1/SUL2 system that senses and transports sulfate (*Kankipati et al., 2015*). Moreover, there is substantial evidence from higher eukaryotes indicating the presence of transporters that exhibit an extra role in nutrient-sensing (*Hyde et al., 2007*; *Pérez-Torras et al., 2013*; *Walch-Liu and Forde, 2008*). Further investigation is required to examine the impact of inorganic nutrients transporters on lifespan and the underlying mechanisms. Considering the crucial role of sulfur-containing substances in regulating lifespan, such as hydrogen sulfide and methionine (*Choi et al., 2019*), the sulfate transporters can serve as a good model for such investigation.

Sulfur is widely recognized as a crucial nutrient that exerts significant regulatory influences on cellular metabolism and proliferation (*Maw, 1963*). Sulfur is mostly absorbed and stored within cells in the form of sulfate, facilitated by specialized membrane transporters (*Cherest et al., 1997*). In yeast, the process of enzymatic reduction converts sulfate to sulfide, which is then integrated into organic molecules by the sulfate assimilation pathway (SAP) (*Thomas and Surdin-Kerjan, 1997*). This pathway involves the utilization of two high-affinity transporters SUL1 and SUL2 (*Khurana et al., 2000*). Both SUL1 and SUL2 have been identified as transceptors (*Kankipati et al., 2015*), which can sense sulfur signals and exhibit a robust response to sulfur deprivation by upregulating sulfate transport. Additionally, they play a crucial role as mediators in the activation of the protein kinase A (PKA) pathway in response to nutrient stimuli, thereby governing various cellular processes (*Thevelein and de Winde, 1999*).

In this study, we examined the regulatory mechanism of SUL1 in relation to yeast longevity. We found that SUL1, instead of SUL2 deletion, significantly extends yeast replicative lifespan (RLS), and the increase of lifespan resulting from the deletion of SUL1 is not contingent upon sulfate transport. The SUL1 mutant strain displayed several traits indicative of decreased PKA activity, including accumulation of stress-protective carbohydrates, increase of MSN2 nuclear localization and upregulation of stress response genes, and downregulation of ribosomal gene expression. We show that lifespan extension depends on the stimulation of MSN2 transcriptional activity and autophagy. Our study provides an example where downstream signaling of nutrient instead of the nutrient itself influences lifespan, which may be general for other nutrient transceptors. Our findings also support the PKA pathway's important role for lifespan regulation.

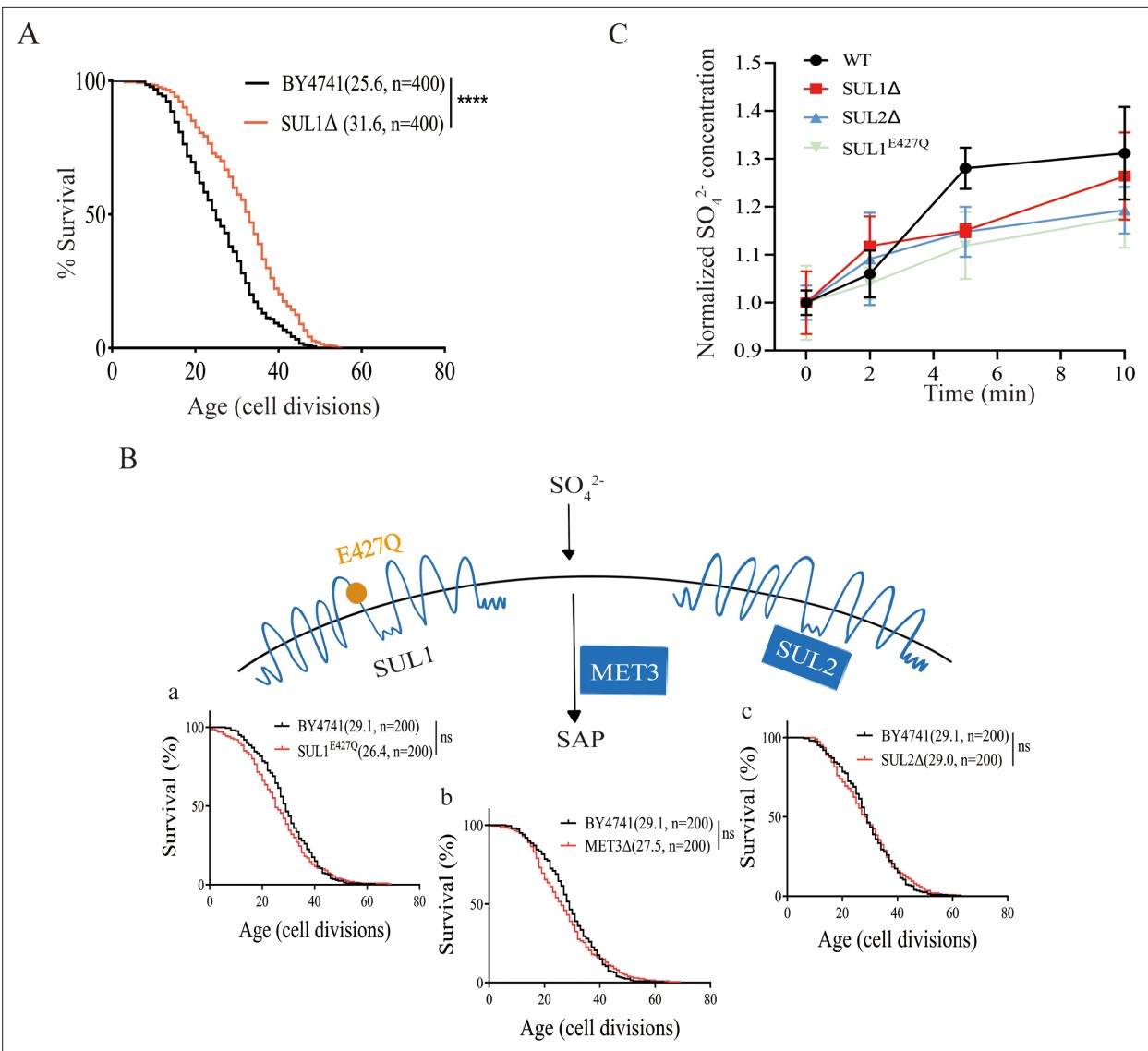

**Figure 1.** The lifespan extension of SUL1Δ mutant is not caused by changes in sulfate transport/metabolism. (**A**) Deletion of the SUL1 gene significantly extended the replicative lifespan of the yeast *Saccharomyces cerevisiae*. Numbers in parentheses indicate the average lifespan and the number of cells measured. ****$p<0.0001$. (**B**) Lifespan is not altered by three targeted genetic interventions that change sulfate transport/metabolism: (a) mutation of the amino acid residue of SUL1 (E427Q) that abolishes the activity of sulfate transport 1; (b) inactivation of MET3, the key enzyme of SAP; (c) deletion of SUL2 (a homolog of SUL1). Survival curves for the WT and SUL1$^{E427Q}$, MET3Δ, or SUL2Δ strains are shown. ns: not significant. (**C**) Time-dependent variations in sulfate ion uptake were assessed in wild-type and mutant strains. The wild-type (WT, black circles), SUL1Δ (red squares), SUL2Δ (blue triangles), and SUL1$^{E427Q}$ (green diamonds) strains were evaluated at 0, 2, 5, and 10 min following stimulation with 3 mM Na$_2$SO$_4$. The Y-axis illustrates the normalized intracellular concentration of sulfate ions. The data points represent the mean values of the ratio to the initial concentration (mg/kg), while the error bars denote the standard deviation of three different experiments.

## Results

### SUL1 deletion increases yeast RLS in a sulfate metabolism-independent way

In budding yeast, SUL1 and SUL2 serve as two plasma membrane transporters responsible for regulating the import of extracellular sulfate. The SUL1 functions as a transporter with a high affinity for sulfate (*Cherest et al., 1997*). Sulfate is converted into various sulfur-containing substances, including methionine, cysteine, glutathione, and S-adenosylmethionine through the SAP (*Marzluf, 1997*). SUL2 is the homolog of SUL1 and may function as a redundant sulfate transporter. We first measured the RLS of SUL1 deletion and found that the mutant has a significantly extended lifespan (*Figure 1A*).

We then seek to understand the mechanism underlying the lifespan extension. Our initial guess was SUL1 deletion decreases the sulfate transport, limiting the availability of sulfur, thus extending lifespan through a mechanism similar to methionine (which also contains sulfur) restriction.

In order to investigate the influence of sulfur metabolism on the extension of lifespan in the SUL1 knockout strain, we generated three strains that modulate the SAP at various levels. These strains include a SUL2 knockout strain, a site-directed mutant strain SUL1$^{E427Q}$, which has been previously demonstrated to disrupt sulfate transport (*Kankipati et al., 2015*), and a MET3 knockout strain that specifically affects a key enzyme in the SAP (*Thomas and Surdin-Kerjan, 1997*). We found that inhibition of the sulfur-transporting function of SUL1 (SUL1$^{E427Q}$) does not extend lifespan. In addition, neither the depletion of SUL2-dependent sulfate transport nor the disruption of SAP through MET3 deletion leads to an increase in the RLS of yeast cells (*Figure 1B*). We then compared the rapid uptake of sulfur following sulfate stimulation in WT, SUL1Δ, SUL2Δ, and SUL1$^{E427Q}$ to evaluate the impact of sulfur transport ability on RLS. Compared with the WT, SUL1Δ, SUL2Δ, and SUL1$^{E427Q}$ all exhibited a delayed sulfur uptake process, but there was no significant difference in the final concentration of intracellular sulfur ion in all groups, and the kinetics of sulfur uptake in SUL1Δ and SUL1$^{E427Q}$ exhibit comparable patterns (*Figure 1C*). These findings indicate that the extended lifespan associated with the deletion of the SUL1 gene is not attributable to alterations in intracellular sulfur ion levels.

## SUL1 deletion induces global transcriptional changes indicative of conserved longevity promotion mechanisms

To gain a deeper understanding of the mechanisms that contribute to the extension of lifespan in SUL1Δ strains, we conducted a comprehensive analysis of the global gene expression profiles of the SUL1Δ strains in comparison with the wild-type strains through RNA-seq. We observed a wide range of changes in the transcript levels of genes in strains lacking SUL1 (*Figure 2A*). A total of 182/57 genes were upregulated/downregulated ($|Log_2 FC|>0.5$) (*Figure 2A*). Genes upregulated are significantly enriched for several biological processes (BP), such as transposition, RNA-mediated regulation, ascospore wall assembly, stress response, protein catabolic process, and carbohydrate metabolism (*Figure 2B and C*, left). On the other hand, genes downregulated are significantly enriched for cellular amino acid biosynthesis, metabolic processes, and ribosome biogenesis (*Figure 2B and C*, right).

The increase in stress response and production of carbohydrates such as trehalose are considered beneficial factors for promoting longevity (*Zhang and Cao, 2017*). Additionally, the suppression of protein biogenesis and ribosomal function is also known to extend lifespan (*McCormick et al., 2015*; *Steyfkens et al., 2018*). Our analysis of gene expression suggests that the deletion of the SUL1 gene triggers a universal, conserved anti-aging mechanism, resulting in an increased lifespan. In contrast, the stress responses that promote longevity and the elevated production of carbohydrates, such as trehalose, are diminished in both the SUL2Δ strains and the SUL1$^{E427Q}$ strains. This observation further supports the genetic mechanism underlying the longevity phenotype, which is exclusively observed in the SUL1Δ strains (*Figure 2—figure supplement 1*).

We observed considerable upregulation of several stress response genes, specifically those controlled by MSN2, in the SUL1 deletion mutant. These genes are involved in the synthesis of trehalose and glycogen, such as TPS1, TPS2, GSY2, and others (*Figure 2C*, left). To identify potential transcriptional regulators, we performed association analysis between TFs and differentially expressed genes (DEGs) (see 'Materials and methods'). We observed that a number of upregulated DEGs in the SUL1 mutant are associated with six stress-related transcription factors, namely SUA7, MSN2, IXR1, FKH1, FKH2, and SFP1 (*Figure 2D*). Interestingly, besides MSN2, which is a well-known general stress response regulator (*Gasch et al., 2000*). We also identified IXR1, a transcriptional repressor that regulates hypoxic genes during normoxia, suggesting that SUL1 deletion induces a gene expression program similar to diauxic shift, a phenomenon also observed in a constitutively active MSN2 mutant that mimics lower PKA activity (*Pfanzagl et al., 2018*).

## SUL1 deletion inhibits the PKA activity and promotes the nuclear translocation of MSN2

Based on our observations, it is evident that the lifespan extension in the SUL1 mutant does not depend on reduced sulfate transport. It has been reported that SUL1 exhibits the ability to react to environmental stress via signaling through the PKA pathway (*Kankipati et al., 2015*). Under conditions

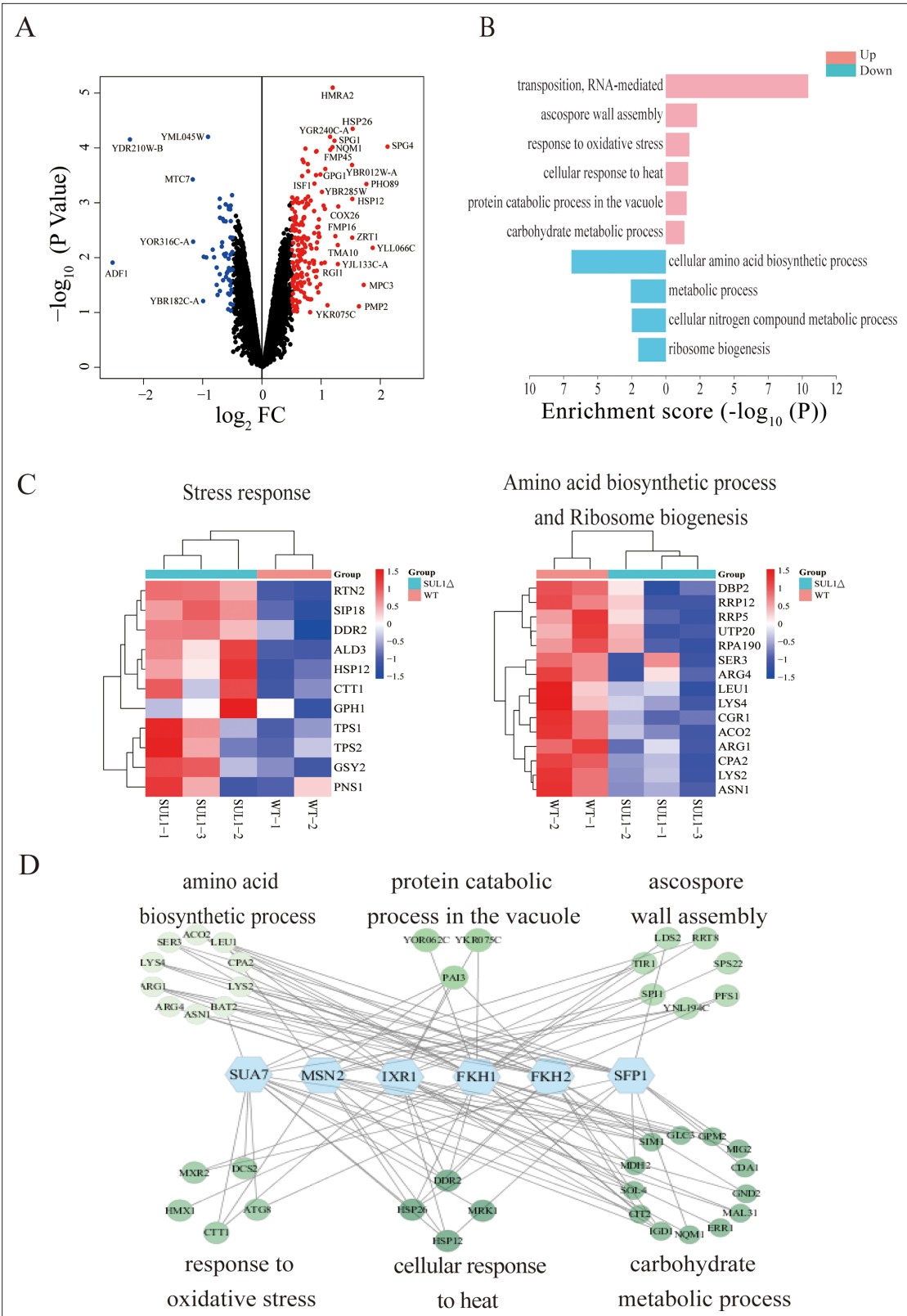

**Figure 2.** Common longevity pathways may contribute to the replicative lifespan (RLS) extension of SUL1 deletion mutant. (**A**) A volcano plot illustrating the differentially expressed genes (DEGs) between the SUL1Δ and WT strains. Log10 of the p-values plotted against the Log$_2$ FC of the fragments per kilobase million (FPKM). (**B**) Enrichment analysis of biological processes associated with the DEGs identified between the SUL1Δ and WT strains. Upregulated genes (p<0.1, Log$_2$ FC>0.5) and downregulated genes (p<0.1, Log$_2$ FC<–0.5) are included in this analysis. (**C**) Heatmaps showing changes

*Figure 2 continued on next page*

*Figure 2 continued*

of stress response (left) and amino acid biosynthetic and ribosome biogenesis genes (right). (**D**) Association analysis of the potential transcription factors and the DEGs in the enriched biological processes.

The online version of this article includes the following figure supplement(s) for figure 2:

**Figure supplement 1.** Transcriptome sequencing analysis was assessed in SUL2Δ VS. WT and SUL$^{E427Q}$ VS. WT.

of nutrient starvation, yeast cells display traits associated with decreased activity of PKA, including the accumulation of stress-protective carbohydrates such as glycogen and trehalose, enhanced resistance to stress, increased expression of stress response genes, and decreased expression of ribosomal genes (*Kankipati et al., 2015*). Additionally, the transcription factors MSN2/MSN4, which are involved in stress resistance, are suppressed by PKA activity. The RNA profile data obtained in our study suggest that the deletion of the SUL1 gene is associated with a decrease in PKA activity and an increase in the expression of genes targeted by MSN2. Thus, to understand the downstream mechanisms for the lifespan extension by SUL1 deletion, we turned our attention to the PKA signaling pathway and its influence on the transcriptional control of MSN2.

The downregulation of the PKA pathway occurs in yeast cells when they are deprived of a crucial nutrient (*Thevelein and de Winde, 1999*). This downregulation leads to adaptive stress protection and the accumulation of trehalose and glycogen (*Lillie and Pringle, 1980*). Consequently, trehalose and glycogen serve as a practical indicator for evaluating the activity of the PKA pathway in vivo (*Schepers et al., 2012*).

We found that the mRNA levels of TPS1 (Trehalose-6-P synthase) and stress-related genes were notably elevated in the SUL1 deletion mutant (*Figure 3A*). Additionally, the concentrations of trehalose and glycogen were significantly increased, indicating a decrease in PKA activity due to SUL1 deletion (*Figure 3B*).

We then further investigated the downstream signal of PKA by examining the expression and the nuclear translocation of MSN2. We constructed an endogenous EGFP-labeled MSN2 in both wild-type and SUL1Δ strains. These strains were then cultured in log phase and subjected to DAPI staining to assess nuclear localization (*Figure 3C*). The ratio of fluorescence intensity (FI) in the nucleus vs. that in the whole cell was then calculated for each individual cell. We observed that the SUL1Δ strain exhibited significantly increased nuclear accumulation of MSN2-EGFP protein compared to the wild-type strain (*Figure 3D*).

To examine the potential changes in the amount of MSN2 protein during the aging process, we utilized a microfluidic device to dynamically monitor the expression of EGFP-tagged MSN2 in individual cells, in both the SUL1Δ and the wild-type strains (*Figure 3E*). We did not observe notable changes in the total amount of MSN2 protein in individual cells in the SUL1Δ strain compared to the wild-type strain (*Figure 3—figure supplement 1*).

However, we did observe a significant increase in nuclear localization of MSN2 over the whole lifespan in the SUL1Δ strain relative to the wild-type strain (*Figure 3F and G*). The SUL1 mutant exhibited a more pronounced change in the nuclear/cytoplasmic cell MSN2 ratio as the cells age (*Figure 3F*). It is worth noting that in the early and middle life stages, specifically prior to the 17th generation, there was a notable difference in the nuclear/cytoplasmic cell MSN2 ratios observed between the two strains. This difference became less significant for the cells passing 17 generations (*Figure 3G*), possibly due to a smaller number of cells that are still alive (25 in the mutant strain and 10 in the wild-type strain) (*Figure 3F*).

In summary, our findings indicate that the deletion of SUL1 results in a decrease in PKA activity and an increase in the nuclear localization of MSN2 (moving from cytoplasm to nucleus), consequently upregulating stress response genes.

## Deletion of SUL1 promotes cellular autophagy

Autophagy is a universal BP that is required for the lifespan-extending effect of various genetic and environmental perturbations (*Sampaio-Marques et al., 2014*). In light of the downregulation of the PKA signal in the SUL1 mutant and the upregulation of the vast majority of autophagy genes identified by the transcriptomic analysis (*Figure 4A*), we investigated the effect of SUL1 deletion on autophagy. We decided to focus on ATG8, initially identified as an essential component of the

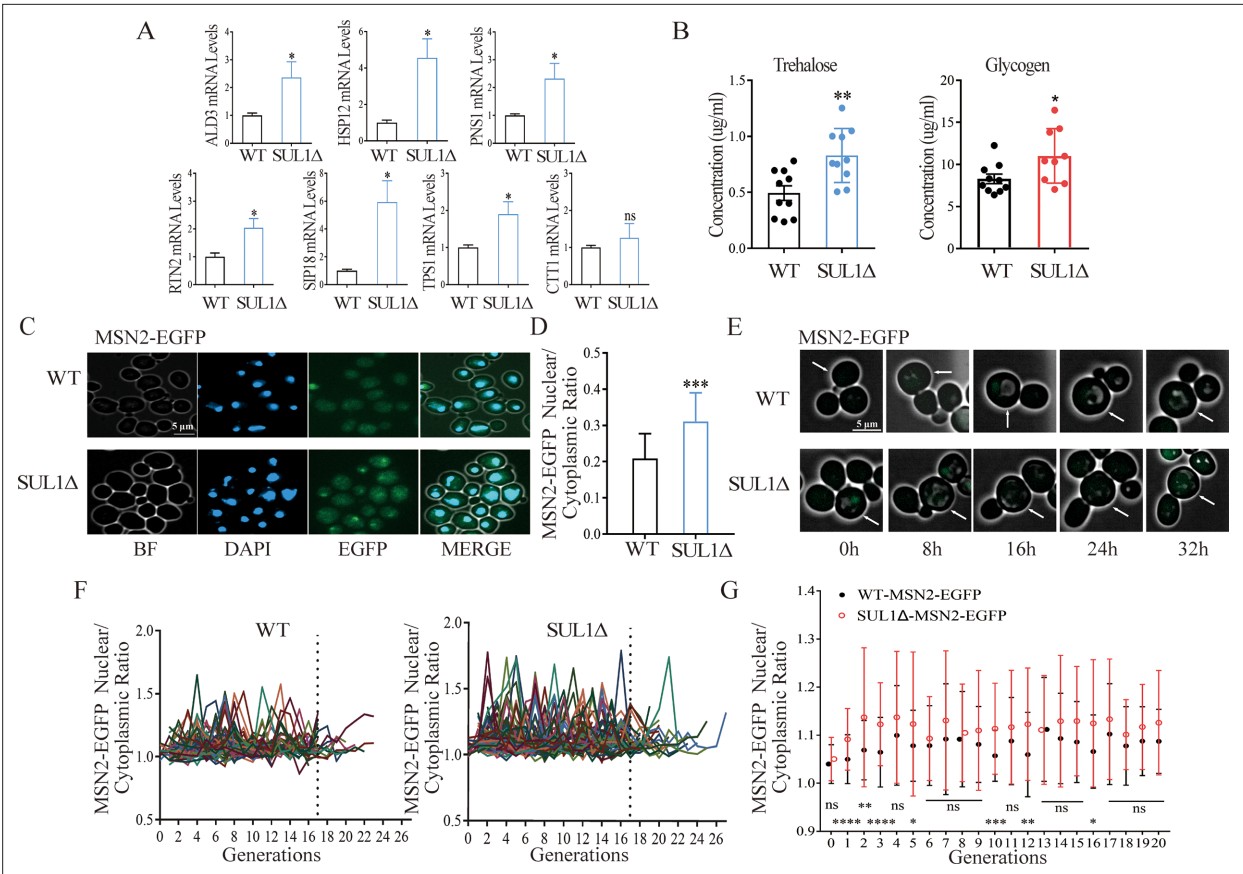

**Figure 3.** SUL1 deletion inhibits the PKA pathway and increases the translocation of MSN2 into the nucleus. (**A**) The mRNA levels of several stress response genes and trehalose synthesis. ns: not significant; *p<0.05. (**B**) The concentrations of trehalose and glycogen in WT and SUL1Δ strains. *p<0.05; **p<0.01. (**C**) Representative images of EGFP-labeled endogenous MSN2 in WT and SUL1Δ strains during the exponential growth phase. BF: bright field. Scale bars: 5 μm. (**D**) The ratio of the mean fluorescence intensity of MSN2-EGFP in the nucleus vs. that of the total cell. Bars represent mean ± SD, n=100. ***p<0.001. (**E**) Representative time-lapse images of MSN2-EGFP in WT and SUL1Δ strains. White arrows represent tracking cells. Scale bars: 5 μm. (**F**) The normalized nuclear/cytoplasmic fluorescence intensity ratio of MSN2-EGFP as a function of age in the WT and SUL1Δ strains (number of cells WT: n=80; SUL1Δ: n=80). The dashed lines represent the nuclear/cytoplasmic ratio of MSN2-EGFP before and after the 17th generation. (**G**) Comparison of the nuclear/cytoplasmic mean fluorescence intensity ratio of MSN2-EGFP as a function of age in WT and SUL1Δ strains. Bars represent mean ± SD, n=80. ns: not significant; *p<0.05; **p<0.01; ***p<0.001; ****p<0.0001.

The online version of this article includes the following figure supplement(s) for figure 3:

**Figure supplement 1.** The expression of MSN2 protein remains stable during aging in the SUL1Δ strain.

**Figure supplement 2.** The nuclear translocation of MSN2 was significantly augmented upon GR stimulation.

**Figure supplement 3.** SUL1 deletion did not promote MSN4 translocation to the nucleus.

autophagy machinery (*Ichimura et al., 2000*; *Paz et al., 2000*). We generated EGFP-labeled ATG8 in both wild-type and SUL1Δ strains. Using a microfluidic device to monitor the dynamic expression of EGFP-tagged ATG8 in individual cells in the SUL1Δ and the wild-type strains, we found that ATG8 levels steadily increase with age in the SUL1Δ strains, but not in the wild-type strain (*Figure 4B and C*). Additionally, a statistically significant difference in ATG8 expression was observed between SUL1Δ and wild-type strains during the middle and late stages of life (*Figure 4D*).

To further examine the difference in autophagy response between the SUL1Δ and the wild-type strains, we introduced glucose restriction as a stress stimulus and dynamically monitored alterations in ATG8 expression before (glucose 2%) and after (glucose 0.05%) the media switch (*Figure 4E*). Notably, a more pronounced upregulation of ATG8 expression was observed following glucose restriction in the SUL1Δ strain (*Figure 4F*).

In summary, the SUL1Δ strain has elevated expression of autophagy genes and is better at upregulating autophagy in response to aging and nutrient depletion.

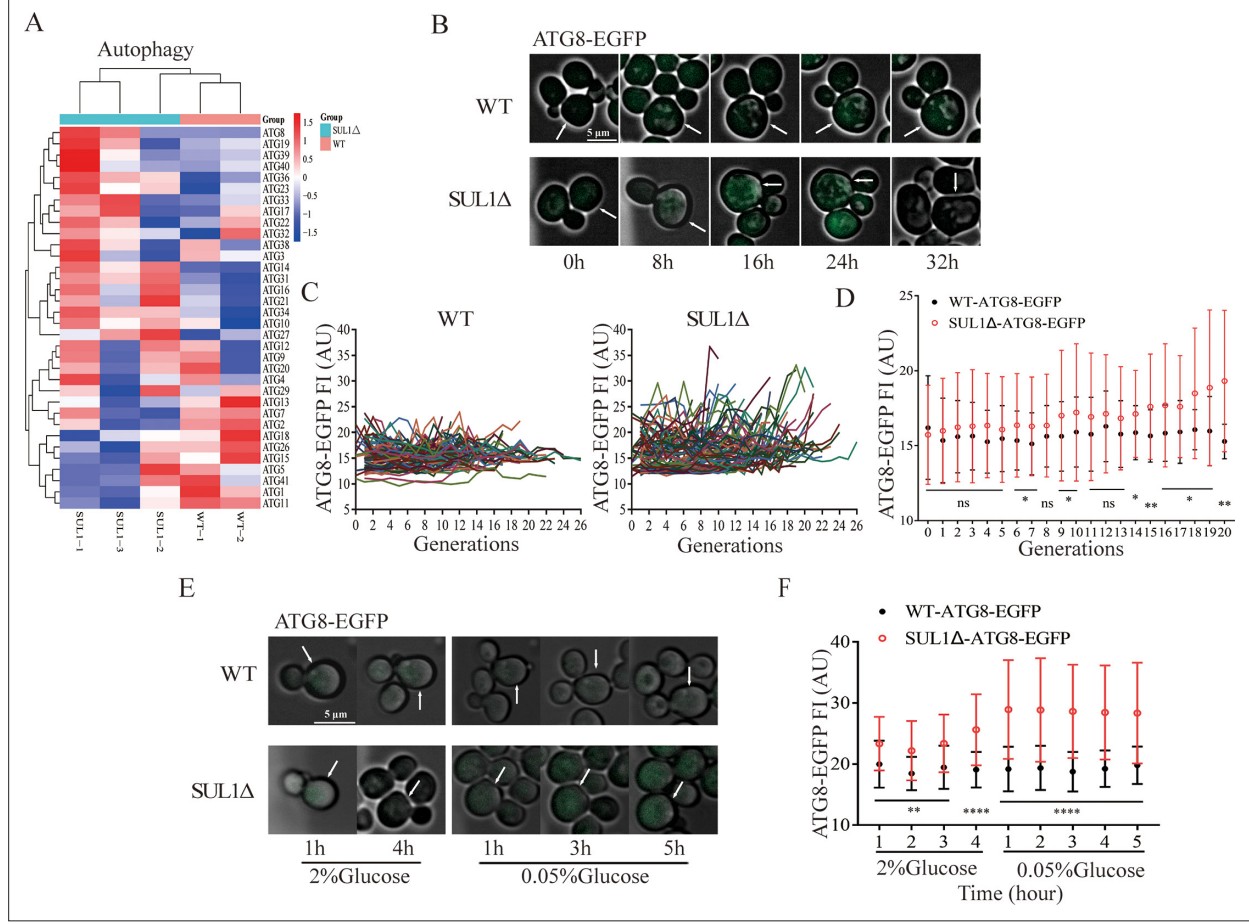

**Figure 4.** SUL1 deletion raises the cellular autophagy level. (**A**) The heatmap vividly showcases the alterations in autophagy-related genes between wild-type (WT) and SUL1Δ strains. (**B**) Representative time-lapse images of ATG8-EGFP in WT and SUL1Δ strains reveal distinct patterns. White arrows represent tracking cells. Scale bars: 5 μm. (**C**) The normalized fluorescence intensity of ATG8-EGFP as a function of age in WT and the SUL1Δ strains, with each colored curve representing a single cell. (Number of cells: WT n=80; SUL1Δ n=80). (**D**) The distribution of the fluorescence intensity of ATG8-EGFP as a function of age in WT and SUL1Δ strains. Bars represent mean ± SD, number of cells n=80. ns: not significant; *p<0.05; **p<0.01. (**E**) Representative time-lapse images of ATG8-EGFP in WT and SUL1Δ strains grown in complete synthetic medium (2% glucose) or in glucose restriction medium (0.05% glucose) at the indicated times. White arrows represent tracking cells. Scale bars: 5 μm. (**F**) The distribution of the fluorescence intensity of ATG8-EGFP in WT and SUL1Δ strains grown in complete synthetic medium (2% glucose) or in glucose restriction medium (0.05% glucose) at the indicated times. Bars represent mean ± SD; WT: n=35; SUL1Δ: n=44. **p<0.01; ****p<0.0001.

## MSN2 and ATG8 are required for the lifespan extension by SUL1 deletion

Given the observation of the elevated nuclear MSN2 level and a stronger ATG8 response during aging in SULΔ strain, we further analyzed whether MNS2 and ATG8 are required for the lifespan extension by SUL1 deletion. We compared the RLS of the single mutants MSN2Δ, ATG8Δ, SUL1Δ, and the double mutants SUL1Δ/ MSN2Δ, and SUL1Δ/ ATG8Δ. We found that the deletion of MSN2 or ATG8 alone neither increases nor decreases lifespan. However, deletion of MSN2 (*Figure 5A*) or ATG8 (*Figure 5B*) in the SUL1Δ background partially abolishes the lifespan extension of SUL1 deletion, arguing that the lifespan extension by SUL1 deletion is at least partially mediated through increased stress response (MSN2) and elevated autophagy (ATG8).

Our findings suggest a mechanistic model in which the PKA pathway plays a pivotal role in regulating lifespan extension in response to SUL1 knockdown. The deletion of SUL1 decreases the activity of the PKA signaling pathway, promoting the accumulation of protective trehalose. By inhibiting ribosome biogenesis, bolstering autophagic capacity, and promoting the translocation of MSN2 and consequently increasing stress response, SUL1Δ cells achieve enhanced adaptive stress resistance and increased lifespan (*Figure 5C*).

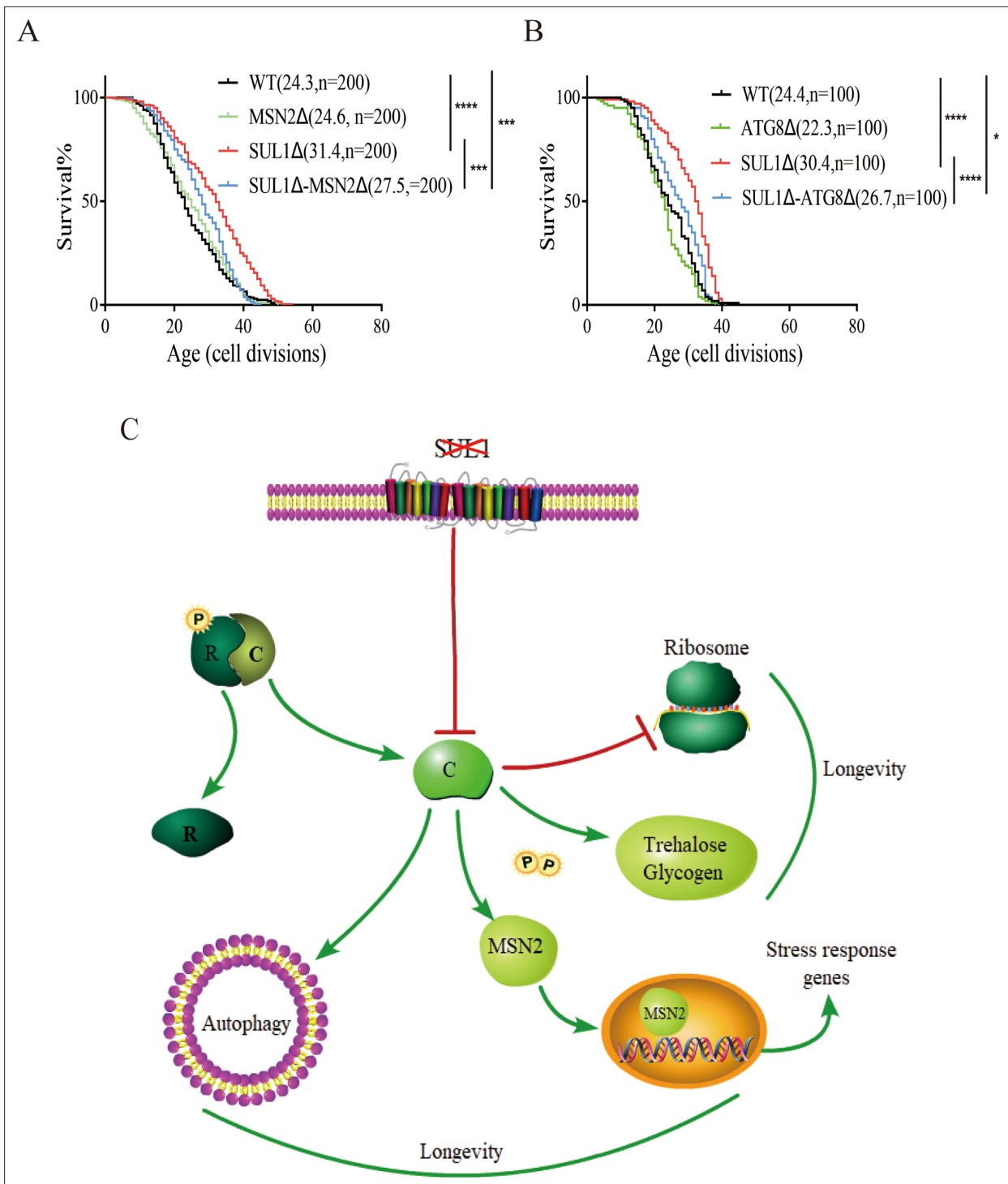

**Figure 5.** The effect of SUL1 deletion on longevity is partially mediated by MSN2 and ATG8. (**A, B**) Replicative lifespan of MSN2 and ATG8 deletion mutants in WT and SUL1Δ strains. The median lifespan and counted cell number are displayed on the graph. (**C**) A schematic illustrating a mechanistic model of how the deletion of SUL1 extends lifespan. SUL1 deletion leads to decreased PKA activity, resulting in increased nuclear translocation of MSN2 (and consequently increased general stress response), autophagy, trehalose, and decreased ribosome biogenesis. The cumulative impact of these downstream effects collectively contributes to the extension of lifespan. R: PKA regulatory subunit; C: PKA catalytic subunit.

## Discussion

Nutrient transceptors are a class of transporters that have the dual function of transporting nutrients as well as sensing and transducing the signal. They can regulate intracellular signaling cascades that exert a pervasive influence on cellular processes, including autophagy, mRNA and ribosome biogenesis, protein synthesis, glucose metabolism, nucleotide and lipid metabolism, proteasomal activity, and stress tolerance (*López-Otín et al., 2023*). Several nutrient-sensing networks have been shown to be associated with longevity in diverse animal models (*Acosta-Rodríguez et al., 2022*; *Duran-Ortiz et al., 2021*; *Spadaro et al., 2022*). In this study, we provide strong evidence that genetic knockout of SUL1, a sulfate-sensing transceptor, extends the RLS of *Saccharomyces cerevisiae* by downregulating PKA signaling to modulate carbohydrate storage, autophagy, ribosome biogenesis, and nuclear translocation of transcription factor MSN2 for general stress response.

Sulfate is transported into the cell of yeast by two high-affinity H-symporters, SUL1 and SUL2 (*Breton and Surdin-Kerjan, 1977*; *Cherest et al., 1997*), where it is reduced to sulfide and homocysteine via the SAP. These compounds are subsequently incorporated into sulfur-containing molecules, including methionine, cysteine, glutathione, and S-adenosylmethionine (*Marzluf, 1997*). These sulfur-containing molecules function as intracellular methyl donors and can modulate the lifespan of various species (*Gu et al., 2017*; *Obata and Miura, 2015*; *Walvekar et al., 2018*). Our study revealed, however, that the downregulation of the high-affinity transceptor SUL2 did not result in an increase in lifespan. Our findings indicate that specific mutations designed to eliminate the sulfate transporter activity of SUL1, while maintaining the signaling capabilities, did not result in an increased lifespan. Furthermore, the disruption of a crucial enzyme in the sulfur assimilation pathway also failed to extend lifespan.

Our study provided an interesting example where lifespan is regulated by the signal transduction downstream of nutrient sensing, instead of the nutrient itself. This might be more general for other nutrient transceptors and is worth further investigation. There is also an interesting parallel in multicellular organisms that the lifespan extension by caloric restriction can be abolished by smelling the food (*Zhang et al., 2021*), highlighting the importance of downstream signaling in regulating lifespan.

Deprivation of a single essential nutrient in a fermentative medium decreases PKA activity (*Thevelein and de Winde, 1999*), leading to the accumulation of reserve and stress protective carbohydrates, glycogen, and trehalose (*Lillie and Pringle, 1980*). PKA plays a crucial role in nutrient regulation of yeast growth and stress tolerance (*Thevelein and de Winde, 1999*). In our transcriptome analysis, the DEGs in SUL1 knockout strain are enriched in BP, including RNA-mediated transposition, protein catabolism, carbohydrate metabolism, stress response, cellular amino acid biosynthesis, and ribosome biogenesis. While in the SUL2 deletion strain and SUL1$^{E427Q}$ mutant strain, the function of DEGs was highly similar (Figure S1), and many enriched categories of them showed opposite trends in the SUL1 deletion strain. For example, the carbohydrate metabolic process-related genes were upregulated in SUL1Δ, while downregulated in SUL2Δ and SUL1$^{E427Q}$ mutant (Figure S1 and *Figure 2B*), indicating SUL1 deletion results in significant PKA signaling change, instead of the interruption of sulfur transport, which contributes to RLS extension. More intriguingly, we found the expression of SUL1 mRNA was significantly increased in the SUL1$^{E427Q}$ mutant (mean FPKM of SUL1 in SUL1$^{E427Q}$ mutant and WT were 37.13 and 21.36, respectively), suggesting that the abolishment of SUL1 sulfur transport function compensatively upregulated the expression of SUL1. But the mechanism of these transposition and metabolic processes demands further exploration.

We identified six transcription factors as important regulators of DEGs in SUL1Δ through a computational analysis: SUA7, which is required for transcription initiation (*Pinto et al., 1994*) IXR1, which regulates hypoxic genes under normoxia conditions *Brown et al., 1993*; FKH1 and FKH2, which regulate the CLB2 gene cluster during the G2/M phase of the mitotic cell cycle (*Bähler, 2005*; *Jorgensen and Tyers, 2000*) SFP1, which regulates ribosomal biogenesis (*Blumberg and Silver, 1991*) and MSN2, which orchestrates the general stress response (*Schmitt and McEntee, 1996*).

In consideration of the diminished activity of the PKA signaling, we specifically focused on MSN2 as a downstream transcription factor of PKA and observed a unique dynamic pattern of nuclear localization in the SUL1Δ strain. Regarding the transcriptional activity of MSN2, the SUL1 mutant exhibited a more robust response to both acute and chronic stimulation. The administration of glucose limitation stimulus, which simulates an acute stress in yeast, resulted in a significant enhancement of MSN2 nuclear entry (*Figure 3—figure supplement 2*). Compared to the wild type, the translocation of

MSN2 into the nucleus was more pronounced in the SUL1Δ strain as cells age, indicating that SUL1 is more sensitive to chronic stresses such as aging-related damage.

In addition to MSN2, MSN4 was initially identified as a PKA-controlled transcription factor that regulates stress response and tolerance genes downstream (*Görner et al., 1998*; *Martínez-Pastor et al., 1996*). MSN2 and MSN4 regulated the transcription of numerous identical downstream targets by binding to the same stress response element, 5'-CCCCT-3' (*Mager and De Kruijff, 1995*). In this study, we also analyzed in-depth the nuclear translocation of MSN4 to determine its response to stresses. Interestingly, compared to the wild type, no significant MSN4 nuclear translocation alteration was observed in SUL1Δ as a result of aging (*Figure 3—figure supplement 3*), indicating that the regulation of stress response capacity of SUL1Δ is MSN2-dependent rather than MSN4-dependent.

In addition, PKA signaling regulates the initiation of the autophagy pathway (*Budovskaya et al., 2004*), and ATG8 is recruited to an expanding structure and is required for the completion of autophagosome formation (*Xie et al., 2008*). Our study revealed that autophagy associated with ATG8 contributed to the lifespan extension of the SUL1Δ strain.

In conclusion, our findings demonstrate that downregulation of SUL1 inhibits the activity of the nutrient-sensing PKA signaling pathway. This regulatory mechanism is responsible for modulating several BP, ultimately leading to an increase in the RLS of yeast. Since the PKA pathway and the associated nutrient response are highly conserved across different organisms (*Kankipati et al., 2015*), our study may have significant implications for the regulation of lifespan in more complex organisms. Further research is needed to clarify the regulatory mechanisms that lead to the weakened PKA signaling pathway following SUL1 knockdown, uncover additional BP that contribute to the lifespan extension, and investigate the complex interactions between the sulfur signaling pathway and other essential nutrient pathways.

## Materials and methods

### Yeast strains, plasmids, and media

All *S. cerevisiae* strains used in this study are in the BY4741 (*MATa his3Δ1 leu2Δ0 met15Δ0 ura3Δ0*) background and are listed in *Supplementary file 1*.

Knockout strains were generated by transforming with a PCR product encoding the kanamycin-resistance (KanR) cassette or HisMX6 cassette to replace the ORF of interest. EGFP-labeled strains were generated by inserting an EGFP-HisMX6 cassette amplified from the pYM-N18-GFP-HisMX6 plasmid into the C-terminal of the target gene. The generation of site-directed mutagenesis strains was achieved by cloning the mutated sequence, synthesized through bridge PCR, into the pUC57 vector downstream of the promoter, utilizing the restriction sites XbaI/SalI. PCR reactions were conducted with Q5 high-fidelity DNA polymerase. Following this, the mutated expression construct was inserted into the yeast expression vector pYM-N18 (HisMX6). All the used primers are provided in *Supplementary file 2*.

The yeast strains were cultivated in YPD complete medium containing 2% glucose, 2% peptone, 1% yeast extract, or synthetic complete medium supplemented with 0.03% essential amino acids and 2% glucose. The glucose restriction media contained the same components as the complete medium except for a reduced concentration of glucose at 0.05%. Yeast strains were grown in selective minimal media at 30°C and shaken at 250–300 rpm.

### Replicative lifespan analysis

The strains were pre-cultured overnight on YPD plates. Lifespan analyses were conducted using micromanipulation, following the previously described protocol (*Lin et al., 2000*). All micromanipulation dissections were performed at laboratory temperature.

### Quantitative determination of intracellular sulfate concentration

Overnight grown yeast cultures were diluted to an $OD_{600}$ of 0.1 and incubated for an additional 3–4 hours to reach the exponential growth phase. Cells were then treated with 3 mM sodium sulfate ($Na_2SO_4$) and harvested at four time points: 0 min, 2 min, 5 min, and 10 min. Immediately after harvesting, the cells were washed with ice-cold PBS buffer and stored at frozen temperatures.

Dilute the frozen cells with PBS buffer to an appropriate concentration and aliquot into centrifuge tubes. Add stabilizer and barium chloride crystals in equal proportions, and immediately vortex until the crystals are completely dissolved. Measure the turbidity of the resulting solution at 420 nm or 480 nm within 15 minutes using a turbidimeter, and calculate the intracellular sulfate ion concentration based on the standard curve.

## RNA-seq and differential expression gene analysis

Yeast in the logarithmic growth phase (~0.5 $OD_{600}$) was harvested, and RNA-seq experiments were conducted following the previously described protocol on BGISEQ-500 (Beijing Genomics Institute). The sequencing data discussed in this publication have been deposited with the NCBI Gene Expression Omnibus and are accessible through the GEO Series accession number GSE296216. The gene expression levels were compared across different samples using fragments per kilobase million (FPKM) values. The $Log_2$ ratio of the FPKM value was used to determine the fold change in gene expression for each sample. The R software was then used to carry out a differential expression analysis between the samples (*Sun et al., 2013*).

## RNA extraction, reverse transcription, and real-time qPCR

The total RNA was extracted using the TRIzol reagent (Invitrogen, #15596026) following the manufacturer's instructions. Subsequently, residual DNA was removed using gDNA wiper mix, and 1 µg of total RNA was reverse-transcribed to cDNA utilizing HiFiScript gDNA Removal RT MasterMix (CWBIO, Jiangsu, China). Quantitative real-time PCR was performed on the 7500 Fast Real-Time PCR System (Applied Biosystems, Foster City, CA, USA) with 2×SuperFast Universal SYBR Master Mix (CWBIO, Jiangsu, China). Actin was used as the internal control for data analysis. The relative expression of each mRNA was calculated using the $2^{-\Delta\Delta CT}$ method. Each measurement consisted of three technical replicates. All primer sequences are included in *Supplementary file 2*.

## Transcriptional factors prediction of DEGs

We utilized the regulator function module of the Saccharomyces Genome Database (SGD) to identify transcription factors associated with DEGs in our transcriptome sequencing data, and ranked their frequency. Transcription factors with higher frequencies were considered more likely to be common regulators of DEGs. The regulatory network of these transcription factors on DEGs was visualized using Cytoscape 3.7.1.

## Glycogen and trehalose quantification

A single clone of yeast was inoculated in 5 ml of YPD medium and incubated overnight at 30°C, 200 rpm. After being diluted to a concentration ($OD_{600}$=0.1), the culture was incubated for 4 hours at the same temperature and speed until it entered the exponential growth phase. The cells were rinsed with 1 ml of sterile ice water and the cell concentration was quantified. $1\times10^8$ cells from the cell suspension were transferred to a 96-well plate, centrifuged, and the supernatant was then removed. After being resuspended in 125 µl of $Na_2CO_3$, the cells were incubated at 95°C for 3 hours while being rotated occasionally. The plates were then cooled to room temperature, and the cell suspension was thoroughly mixed before being equally divided into two new plates. There were three technical replicates for each strain.

For glycogen measurements, each reaction solution was meticulously prepared by combining 188 µl of amyloglucosidase buffer, 62 µl of cell suspension, and 10 µl of freshly prepared *Aspergillus niger* α-amyloglucosidase solution (~70 U/mg) in a sodium acetate buffer (pH = 5.2) at a concentration of 0.2 M. The plates were thoroughly mixed and incubated overnight at 57°C.

For trehalose measurements, each reaction solution was precisely formulated by adding 188 µl of trehalase buffer, 62 µl of cell suspension, and 10 µl of porcine trehalase solution (~0.007 U) diluted threefold with 0.2 M sodium acetate buffer (pH = 5.2). The plates were subsequently incubated overnight at 37°C.

The quantification of glucose released in the preceding procedures was carried out using the Glucose Assay Kit (Sigma-Aldrich, #MAK263) in accordance with the detailed instructions provided by the manufacturer.

## Single-cell time-lapse imaging

The cells were cultured overnight in synthetic complete media containing 2% glucose at 30°C until they reached an optical density of approximately 1.0. Subsequently, the cells were diluted 10-fold and further cultivated in a shaker at 30°C for an additional 4–6 hours before being loaded into the microfluidic device. The microfluidic device was employed to track single cells, following the previously described protocols (*Chen et al., 2020*; *Xie et al., 2012*). The microfluidic chip used for this experiment was generously provided by Prof. Chunxiong Luo from PKU.

## Fluorescence imaging

Yeast cells in the logarithmic growth phase were used to observe the nuclear localization of MSN2-EGFP or MSN4-EGFP. The nuclei were stained with DAPI at a final concentration of 2.5 µg/ml and incubated for 30 min at room temperature. Images were captured using Zeiss Axio Observer 7, and image analysis was conducted using the ImageJ software. In brief, the quantification of protein expression abundance in each cell is determined by subtracting the background intensity from the green fluorescence intensity. In each experiment, a minimum of 100 cells were subjected to analysis.

## Flow cytometry

The EGFP-labeled strains were incubated overnight, followed by being diluted to an $OD_{600}$ equal to 0.01. The cultures were then grown for an additional duration of 2–4 hours until they attained an $OD_{600}$ of approximately 0.05 in a 96-well plate. Subsequently, flow cytometry was employed to quantify the EGFP signals (FITC channel) in each sample at regular hourly intervals. The cellular concentration of EGFP was assessed by normalizing the EGFP signal with cell size using a forward scattering signal (FSC channel) for individual cells. Mean fluorescence intensity was utilized to compare the differences.

## Statistical analysis

The RLS was analyzed using the Kaplan–Meier method, and the survival differences were determined by the log-rank test. Other group differences were evaluated with Student's *t*-test. The rapid sulfur intake experiment was analyzed using repeated measures ANOVA. The statistical analyses were conducted using SPSS 22.0 and GraphPad Prism 6. Statistical significance was considered at p-value <0.05.

## Acknowledgements

This work was supported by Sichuan Science and Technology Program (no. 2022ZYD0076; no. 2023YFS0050), and the funding supported by Medico-Engineering Cooperation Funds from the University of Electronic Science and Technology of China and China West Hospital (grant no. ZYGX2022YGRH018).

# Additional information

## Funding

| Funder | Grant reference number | Author |
|---|---|---|
| Sichuan Science and Technology Program | No. 2022ZYD0076 | Jing Yang |
| Sichuan Science and Technology Program | No. 2023YFS0050 | Jing Yang |
| China West Hospital | Grant No. ZYGX2022YGRH018 | Jing Yang |

The funders had no role in study design, data collection and interpretation, or the decision to submit the work for publication.

## Author contributions
Juan Long, Conceptualization, Methodology, Writing – original draft, Project administration; Meng Ma, Software, Formal analysis; Yuting Chen, Methodology; Bo Gong, Resources, Supervision; Yi Zheng, Hao Li, Visualization, Writing – review and editing; Jing Yang, Supervision, Funding acquisition, Validation, Writing – review and editing

## Author ORCIDs
Juan Long ⓘ https://orcid.org/0009-0003-8567-2331
Yi Zheng ⓘ https://orcid.org/0000-0002-5230-0254
Jing Yang ⓘ https://orcid.org/0000-0002-9991-5010

Reviewer #1 (Public review): https://doi.org/10.7554/eLife.94609.3.sa1
Reviewer #2 (Public review): https://doi.org/10.7554/eLife.94609.3.sa2
Reviewer #3 (Public review): https://doi.org/10.7554/eLife.94609.3.sa3
Author response https://doi.org/10.7554/eLife.94609.3.sa4

## Additional files

### Supplementary files
Supplementary file 1. Strains used in this study, related to experimental procedures.

Supplementary file 2. Primers used in this study, related to experimental procedures.

MDAR checklist

### Data availability
Sequencing data have been deposited in GEO under accession codes GSE296216.

The following dataset was generated:

| Author(s) | Year | Dataset title | Dataset URL | Database and Identifier |
|---|---|---|---|---|
| Juan L, Hao L, Jing Y | 2025 | SUL1 deficiency reduces PKA activity to increase replicative lifespan in yeast *Saccharomyces cerevisiae* | https://www.ncbi.nlm.nih.gov/gds/?term=GSE296216 | NCBI Gene Expression Omnibus, GSE296216 |

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
