## [Editor Report · eLife Assessment]

This study offers a **valuable** contribution to the understanding of how inorganic nutrient transporters, particularly SUL1, influence yeast lifespan through signaling pathways rather than transport functions. The findings suggest a novel link between SUL1 deletion and extended replicative lifespan, supported by transcriptomic and stress–response data. However, the strength of the evidence remains **incomplete**, with key experiments—such as sulfate supplementation tests, functional autophagy validation, and transport assays—either missing or insufficiently described. As a result, while the manuscript presents promising insights, additional work is needed to robustly support its conclusions.

---

## [Referee Report · Reviewer #1 (Public review)]

The manuscript by Long et al. focused on SUL1, a gene encoding a sulfate transporter with signaling roles in yeast. The authors claim that the deletion of SUL1, rather than SUL2 (encoding a similar transporter), extended yeast replicative lifespan independent of sulfate transport. They also show that SUL1 loss-of-function mutants display decreased PKA activity, indicated by stress-protective carbohydrate accumulation, relevant transcription factor relocalization (measured during aging in single cells), and changes in gene expression. Finally, they show that loss of SUL1 increases autophagy, which is consistent with the longer lifespan of these cells. Overall, this is an interesting paper, but additional work should strengthen several conclusions, especially for the role of sulfate transport. Specific points include the following:

What prompted the authors to measure the RLS of sul1 mutants? Prior systematic surveys of RLS in the same strain background (which included the same sul1 deletion strain they used) did not report lifespan extension in sul1 cells (PMID: 26456335).

Cells carrying a mutant Sul1 (E427Q), which was reported to be disrupted in sulfate transport, did not have a longer lifespan (Figure 1), leading them to conclude that "lifespan extension by SUL1 deletion is not caused by decreased sulfate uptake". They would need to measure sulfate uptake in the mutants they test to draw that conclusion firmly.

Related to my previous point, another simple experiment would be to repeat the assays in Figure 1 with exogenous sulfur added to see if the lifespan extension is suppressed.

There needs to be more information in the text or the methods about how they did the enrichment analysis in Figure 2B. P-values are typically insufficient, and adjusted FDR values are reported from standard gene ontology platforms (e.g., PANTHER).

It is somewhat puzzling that relocalization of Msn2 was not seen in very old cells (past the 17th generation), but it was evident in younger cells. The authors could consider another possibility, that it was early and midlife experiences that made those cells live longer. Past that window, loss of Sul1 may have no impact on longevity. A conditional shutoff system to regulate SUL1 expression would be needed to test the above, albeit this is probably beyond the scope of this report.

The connections between glucose restriction, autophagy, and sul1 (Figure 4) could be further tested by measuring the RLS of sul1 cells in glucose-restricted cells. If RLS is further extended by glucose restriction, then whatever effects they see should be independent of glucose restriction.

They made and tested the double (sul1, msn2) mutants, but they should also test the sul1, msn4 combination since Msn4 functions similarly to Msn2.

Comments on revisions:

Overall, this is a somewhat improved manuscript, but some prior concerns about the validity of the conclusions remain unresolved.

---

## [Referee Report · Reviewer #2 (Public review)]

Summary:

In this study, the authors find that deletion of a sulfate transporter in yeast, Sul1, leads to extension of replicative lifespan. They investigate mechanisms underlying this extension, and claim that the effects on longevity can be separated from sulfate transport, and are instead linked to a previously proposed transceptor function of the Sul1 transporter. Through RNA sequencing analysis, the authors find that Sul1 loss triggers activation of several stress response pathways, and conclude that deletion of two pathways, autophagy or Msn2/4, partially prevents lifespan extension in cells lacking Sul1. Overall, while it is well-appreciated that activation of Msn2/4 or autophagy is beneficial for lifespan extension in yeast, the results of this study would add an important new mechanism by which this could achieved, through perceived sulfate starvation. However, as described below, several of the experiments utilized to support the authors conclusion are not experimentally sound, and significant additional experimentation is required to support the authors claims throughout the manuscript.

Strengths:

The major strength of the study is the robust RNA-seq data that identified differentially expressed genes in cells lacking Sul1. This facilitated the authors focus on two of these pathways, autophagy and the Msn2/4 stress response pathway.

Weaknesses:

Several critical experimental flaws need to be addressed by the authors to more rigorously test their hypothesis.

(1) The lifespan assays throughout the manuscript contain inconsistencies in the mean lifespan of the wild type strain, BY4741. For example, in Figure 1A, the lifespan of BY4741 is 24.3, and the extended lifespan of the sul1 mutant is 31. However, although all mutants tested in Figure 1B also have lifespans close to 30 cell divisions, the wild type control is also at 30 divisions in those experiments as well. This is problematic, as it makes it impossible to conclude anything about the lifespan extension of various mutants with the inconsistencies in the wild type lifespan. Additionally, the mutants analyzed in 1B are what the authors use to claim that loss of the transporter does not extend lifespan through sulfate limitation, but instead through a signaling function. Thus, it remains unclear whether loss of sul1 extends lifespan at all, and if it does, whether this is separable from cellular sulfate levels.

(2) While the authors use mutants in Figure 1 that should have differential effects on sulfate levels in cells, the authors need to include experiments to measure sulfate levels in their various mutant cells to draw any conclusions about their data.

(3) Similar to point 2, the authors focused their RNA sequencing analysis on deletion of sul1 and did not include important RNA seq analysis of the specific Sul1 mutation or other mutants in Figure 1B that do not exhibit lifespan extension. The prediction is that they should not see activation of stress response pathways in these mutants as they do not see lifespan extension, but this needs to be tested.

(4) While the RNA-seq data is robust in Figure 2 as well as the follow up quantitative PCR and trehalose/glycogen assays in 2A-B, the follow-up imaging assays for Msn2/4 localization in Figure 2 are not robust and are difficult to interpret. The authors need to include more high-resolution imaging or at least a close up of the cells in Figure 3C.

(5) The autophagy assays utilized in Figure 4 appear to all be done with a C-terminal GFP-tagged Atg8 protein. As C-terminal GFP is removed from Atg8 prior to conjugation to phosphatidylethanolamine, microscopy assays of this reporter cannot be utilized to report on autophagy activity or flux. Instead, the authors need to utilize N-terminally tagged Atg8, which they can monitor for vacuole uptake as an appropriate readout of autophagy levels. As it stands, the authors cannot draw any conclusions about autophagy activity in their studies.

Comments on revisions:

Their autophagy conclusions are weak at best. As was highlighted in the previous review, they need to use an N-terminal Atg8 fusion for these experiments.

---

## [Referee Report · Reviewer #3 (Public review)]

Summary:

In the revised manuscript, Long et al., showed that sul1∆ mutants have extended replicative lifespan in budding yeast. In comparison, other mutants that have sulfate transport deficiency did not show extended lifespan, suggesting SUL1 deletion extends lifespan independently of sulfate intake. The authors then explored the transcriptome of sul1∆ mutants by RNA-seq, which suggests that SUL1 deletion impacts common longevity pathways. Furthermore, the authors characterized how the PKA pathway is affected in sul1∆ mutants: SUL1 deletion promotes the nuclear localization of Msn2, as well as autophagy, indicating down-regulation of the PKA pathway.

Strengths:

This study raised an interesting point that inorganic transporters may impact cellular stress response pathways and affect lifespan. Some of the characterizations on the sul1∆ mutants, including the RNA-seq and MSN2 localization could provide valuable sources for people in related fields. Compared with the previous version, the writing is significantly improved, making the manuscript clearer.

Weaknesses:

Several critical flaws have not been revised. The claims are still not well supported by the data.

(1) The revised manuscript still uses Atg8-EGFP, in which GFP is likely tagging at the C-terminus of Atg8. No strain information was provided for this strain, so it is unclear whether it is N- or C- terminal tagged. As pointed by reviewers of the previous version, C-terminal tagged Atg8 is not functional. As a result, the conclusions on autophagy (Figure 4) is questionable.

(2) The nuclear localization of Msn2 is much more convincing after the authors updated Figure 3C. However, the rest of the microscopy images (e.g. Figure 3E, 4B, 4E) are still of low resolution. Again, I suggest to separate the DIC and GFP channels. It is really hard to tell where is the GFP signal from these figures.

(3) In the Kankipati et al. 2015 paper, which is cited by the authors, SUL1E427Q is incorporated on a pRS316 (URA3) plasmic and expressed in sul1∆sul2∆ mutants. In this manuscript, the authors used SUL1E427Q mutants but did not give detailed information on how this construct is expressed. Is it endogenously mutated, incorporated into somewhere in the genome, or expressed from an extrachromosomal plasmid?

In Figure 1B, they simply used BY4741 as a control for the SUL1E427Q mutant. This makes me thinking they are using a SUL1E427Q endogenous point mutation mutant. If so, the authors may want to include the information about this strain in their Supplementary table. Or if it is expressed from an extra copy on chromosomes or extrachromosomal plasmids, the authors would need to express this construct in sul1∆ mutant. In this case, the authors may want to use sul1∆ and sul1∆+empty vector as controls, instead of BY4741. As the authors mentioned in their rebuttal letter, lifespan experiments vary between each individual trials and are not comparable between different trials. Thus proper controls are essential to make the results convincing.

(4) As suggested by reviewers of the previous version, the authors tested the sulfate uptake in different mutants within 10 minute of Na2SO4 addition (Figure 1B). The authors concluded from the data that wild type takes up sulfate faster than the mutants but they reach similar concentrations at the end point (as fast as 10 minutes). Are all these cells sulfate-starved before the experiment? If not, the experiment might be affected by the basal level of sulfate in each mutants.

---

## [Author Response]

The following is the authors’ response to the original reviews.

**Reviewer #1 (Recommendations for authors):**
(1) Motivation for studying SUL1 in RLS

Considering that the regulation of cellular metabolism in response to nutrient availability is crucial for cell survival and lifespan, and several organic nutrient transporters have also been implicated in the mediation of aging, we believe that transporters of specific nutrients can transduce the signal downstream to control genes responsible for survival. However, the impact of inorganic nutrient transporters, including phosphate and sulfate, on longevity remains largely unexplored. And another work of our group utilized a LASSO model derived from multi-omics data related to yeast aging, identifying SUL1 as a key candidate for regulating lifespan, which aroused our interest.

(2) Discrepancy with prior RLS data (PMID: 26456335)

Previous literature (PMID: 26456335) reported a limited number of experimental cells (n=25), which may have contributed to the observed variability in results. To enhance the reliability of our work, we have expanded the number of experimental cells for the sul1Δ strain to 400 (see Figure 1A). In contrast, the lifespan data for other mutant strains have been increased to 200 (see Figure 1B). This confirms the reproducibility of the lifespan extension observed in the sul1Δ strain.

(3) Mechanistic link between sulfate transport and lifespan

Sulfate absorption assays were performed on the WT, SUL1Δ, SUL2Δ, and SUL1^E427Q^ strains (Figure 1C). Compared to the wild type (WT), the SUL1Δ, SUL2Δ, and SUL1^E427Q^ strains exhibited delayed sulfate intracellular transportation. However, there was no significant difference in the final concentration of intracellular sulfur ions among all groups. This result reinforces our conclusion that the extended lifespan of SUL1Δ is not associated with sulfate transport.

(4) Testing the RLS of SUL1ΔMSN4Δ double mutants

The replicative lifespan data for the SUL1ΔMSN4Δ double mutant were further analyzed (shown in the following supplementary figure). It was observed that the extension of the SUL1Δ lifespan was not rescued by the knockout of MSN4, supporting the hypothesis that MSN2 may serve as the downstream transcription factor responsible for the increased lifespan of SUL1Δ.

**Author response image 1. sa4fig1:** Replicative life span of MSN4 deletion mutants in WT and SUL1Δ strains.

**Reviewer #2 (Recommendations for authors):**
(1) Inconsistent WT lifespan in Figure 1B

All measurements of life expectancy were conducted under controlled conditions (30°C, 2% glucose). The revised Figure 1C illustrates that across three independent experiments (n=200 cells), the average lifespan of wild-type (WT) cells was 29.1 generations, which is comparable to the average lifespan of 25.6 generations reported in Figure 1A after data expansion (n=400 cells). This similarity may be attributed to experimental variability arising from multiple trials; however, it does not compromise the validity of our conclusions.

(2) Sulfate level measurements

Intracellular sulfate levels were measured by quantitatively assessing the sulfate concentrations in wild-type (WT), SUL1Δ, SUL2Δ, and SUL^E427^ cells, as detailed in the methods section (Figure 1C). The results indicated that all mutant strains showed a delayed sulfur uptake process, but there was no significant difference in the final concentration of intracellular sulfur ions in all groups.

(3) RNA-seq for non-lifespan-extending mutants

RNA-seq data for the SUL2Δ and SULE427 mutants can be found in Supplementary Figure 1. These mutants do not exhibit a significant upregulation of stress-response genes, such as HSP12 and TPS1, which reinforces the specificity of the pathways induced by SUL1Δ.

(4) Improved Msn2/4 imaging

Figure 3C and supplementary Figure 4A present high-resolution confocal images (using a 63× objective lens) of cell nuclei labeled with MSN2-GFP and DAPI. The GFP intensity within the nucleus was normalized against the DAPI signal to account for differences in nuclear size.

**Reviewer #3 (Recommendations for authors):**
(1) Nuclear size normalization

The verification data for MSN2 and MSN4 were re-evaluated through DAPI signal normalization. The revised figures are presented in Figure 3C and Supplementary Figure 4A.

(2) Strain nomenclature

All strain names (e.g., SUL1Δ) were updated to follow SGD guidelines.

(3) Grammar and formatting

We have carefully revised the text to improve readability. And the manuscript was proofread by a native English speaker. Citations (e.g., "trehalose (Lillie and Pringle, 1980)") and spacing errors were corrected.

(4) Microscopy resolution

In the revised figures (Figures 3C, 3E, 4B, 4E, Supplementary Figure 3A, 4A, 4C), all fluorescence images are displayed as separate channels (EGFP, DAPI, BF). The scale and arrows have been added to the figure for clarity.